# Growth and Competitive Infection Behaviors of *Bradyrhizobium japonicum* and *Bradyrhizobium elkanii* at Different Temperatures

**Md Hafizur Rahman Hafiz** [1,2], **Ahsanul Salehin** [3] **and Kazuhito Itoh** [1,3,*]

[1] Faculty of Life and Environmental Science, Shimane University, 1060 Nishikawatsu, Matsue 690-8504, Japan; hafizhstu@hotmail.com

[2] Department of Crop Physiology and Ecology, Hajee Mohammad Danesh Science and Technology University, Dinajpur 5200, Bangladesh

[3] The United Graduate School of Agricultural Sciences, Tottori University, 4-101 Koyama-Minami, Tottori 680-8553, Japan; ujanrijvi224@gmail.com

[*] Correspondence: itohkz@life.shimane-u.ac.jp; Tel.: +81-852-32-6521

**Abstract:** Growth and competitive infection behaviors of two sets of *Bradyrhizobium* spp. strains were examined at different temperatures to explain strain-specific soybean nodulation under local climate conditions. Each set consisted of three strains—*B. japonicum* Hh 16-9 (Bj11-1), *B. japonicum* Hh 16-25 (Bj11-2), and *B. elkanii* Hk 16-7 (BeL7); and *B. japonicum* Kh 16-43 (Bj10J-2), *B. japonicum* Kh 16-64 (Bj10J-4), and *B. elkanii* Kh 16-7 (BeL7)—which were isolated from the soybean nodules cultivated in Fukagawa and Miyazaki soils, respectively. The growth of each strain was evaluated in Yeast Mannitol (YM) liquid medium at 15, 20, 25, 30, and 35 °C with shaking at 125 rpm for one week while measuring their $OD_{660}$ daily. In the competitive infection experiment, each set of the strains was inoculated in sterilized vermiculite followed by sowing surface-sterilized soybean seeds, and they were cultivated at 20/18 °C and 30/28 °C in a 16/8 h (day/night) cycle in a phytotron for three weeks, then nodule compositions were determined based on the partial 16S-23R rRNA internal transcribes spacer (ITS) gene sequence of DNA extracted from the nodules. The optimum growth temperatures were at 15–20 °C for all *B. japonicum* strains, while they were at 25–35 °C for all *B. elkanii* strains. In the competitive experiment with the Fukagawa strains, Bj11-1 and BeL7 dominated in the nodules at the low and high temperatures, respectively. In the Miyazaki strains, BjS10J-2 and BeL7 dominated at the low and high temperatures, respectively. It can be assumed that temperature of soil affects rhizobia growth in rhizospheres and could be a reason for the different competitive properties of *B. japonicum* and *B. elkanii* strains at different temperatures. In addition, competitive infection was suggested between the *B. japonicum* strains.

**Keywords:** *Bradyrhizobium japonicum*; *Bradyrhizobium elkanii*; temperature effects; growth; competitive infection; nodule composition

## 1. Introduction

Soybean-nodulating bacteria have distributed worldwide [1,2] and established important symbiotic relationships with host plants to fix atmospheric nitrogen [3]. *Bradyrhizobium japonicum* and *B. elkanii* are reported as the major soybean nodulating rhizobia [4,5] and their nodulation behaviors in the field need to be clarified in relation to environmental conditions because their nodulation and nitrogen fixation are known to be highly dependent on environmental conditions [6]. In previous studies, latitudinal-characteristic nodulation of *B. japonicum* and *B. elkanii* has been reported in Japan [7,8], the United States [9], and Nepal [10], in which *B. japonicum* and *B. elkanii* dominate in soybean nodules in northern and southern regions, respectively. These results suggest that the temperature of the soybean-growing location contributes to the nodule composition of *B. japonicum* and *B. elkanii*.

To elucidate the possible reason, laboratory competitive inoculation experiments have been conducted at different temperatures. Kluson et al. [11] reported that *B. japonicum* strains dominated in nodules at lower temperatures, while *B. elkanii* strains dominated at higher temperature. Suzuki et al. [12] examined the relative population of *B. japonicum* and *B. elkanii* strains in the rhizospheres of soybeans and their nodule compositions at different temperatures and revealed that the *B. japonicum* strain dominated in nodules at lower temperature even though the relative populations of both strains were similar in the rhizosphere, while at higher temperature, the *B. elkanii* strains dominated in nodules due to their larger relative population in the rhizosphere. Shiro et al. [13] reported that the nodule occupancy of *B. elkanii* increased at higher temperatures, whereas that of *B. japonicum* increased at lower temperatures, corresponding to their temperature-dependent *nodC* gene expressions. These results suggest that the temperature-dependent infections and proliferations in soils are possible reasons for the temperature-dependent nodule compositions of rhizobia in the field. However, it has been uncertain which factor, namely, temperature-dependent infection or proliferation in soil, contributes to the temperature-dependent distribution of rhizobia in nodules.

For elucidating which factor is more involved in the soybean nodule composition under local climatic conditions, Hafiz et al. [7] examined the changes in the nodule composition when soil samples were used for soybean cultivation under the different climatic conditions from the original locations, and found that the *B. japonicum* strains nodulated dominantly in the Fukagawa location (temperate continental climate) and the dominancy of *B. japonicum* did not change when soybean was cultivated in the Matsue and Miyazaki locations (humid sub-tropical climate) using the Fukagawa soil. The results suggest that the *B. japonicum* strains proliferated dominantly in the Fukagawa soil leading to their nodule dominancy because *B. elkanii* did not appear in the Matsue and Miyazaki locations. On the other hand, the *B. elkanii* strains dominated in the Miyazaki soil and location while the *B. japonicum* strains dominated when soybean was cultivated in the Fukagawa location using the Miyazaki soil, suggesting that temperature-dependent infection would lead to nodule dominancy of the *B. elkanii* and *B. japonicum* strains in the Miyazaki and Fukagawa locations, respectively.

In addition, in the Fukagawa soil and location, phylogenetic sub-group *B. japonicum* Bj11-1, which was characterized as a slow grower, dominated the nodules compared to another sub-group *B. japonicum* Bj11-2, which was characterized as a fast grower [7], suggesting that infection preference might determine the nodule composition among the *B. japonicum* strains rather than their growth properties. In the Miyazaki soil and location, it was suggested that both *B. japonicum* and *B. elkanii* strains proliferated, and that the species-specific nodule compositions under the different local climatic conditions might be due to the temperature-dependent growth and infection properties of the *Bradyrhizobium* strains [7].

These hypotheses presented in the previous study [7] should be confirmed by in vitro growth and inoculation experiments under the controlled temperatures using the *B. japonicum* and *B. elkanii* strains isolated from the corresponding soils and locations. In this study, we compared growth and infection behaviors at different temperatures of the *B. japonicum* and *B. elkanii* strains isolated from the soybean nodules cultivated in the Fukagawa and Miyazaki soils, and elucidated the reason why the species-specific nodule compositions are present in the Fukagawa and Miyazaki soils and locations.

## 2. Materials and Methods

### 2.1. Effect of Temperature on Growth of Bradyrhizobium spp. in Liquid Culture

The strains used are listed in Table 1. They were isolated from nodules of soybean cultivated in the Fukagawa and Miyazaki soils and study locations in 2016, and selected based on their phylogenetic characteristics based on the 16S rRNA and 16S-23S rRNA internal transcribes spacer (ITS) gene sequences [7].

**Table 1.** *Bradyrhizobium* strains used in this study.

| Strain [a] | Closest 16 rDNA | ITS Group [b] | Accession Number [c] |
|---|---|---|---|
| Hh 16-9 | *B. japonicum* Bj11 | Bj11-1 | LC582854, LC579849 |
| Hh-16-25 | *B. japonicum* Bj11 | Bj11-2 | LC582860, LC579855 |
| Hk 16-7 | *B. elkanii* L7 | BeL7 | LC582891, LC579886 |
| Kh 16-43 | *B. japonicum* S10J | BjS10J-2 | LC582874, LC579869 |
| Kh 16-64 | *B. japonicum* S10J | BjS10J-4 | LC582887, LC579882 |
| Kh 16-7 | *B. elkanii* L7 | BeL7 | LC582901, LC579896 |

[a] The strains were isolated from nodules of soybean cultivated using Fukagawa (H) and Miyazaki (K) soils at Fukagawa (h) and Miyazaki (k) study locations in 2016. The isolates were designated by soil, location, year, and strain number. [b] Group based on gene sequence of 16S-23S rRNA internal transcribes spacer (ITS) region. [c] Gene accession number of 16S rRNA and ITS sequences.

Considering the temperature ranges during the soybean cultivation period in the study locations (Table 2), the temperatures were set at 15 °C (around average daily minimum temperature in the Fukagawa location), 20 °C (around average daily temperature in the Fukagawa location), 25 °C (around average daily maximum and minimum temperatures in the Fukagawa and Miyazaki locations, respectively), 30 °C (around average daily temperature in the Miyazaki location), and 35 °C (around average daily maximum temperature in the Miyazaki location).

**Table 2.** Geographical and climatic characteristics of the study locations in Japan [7].

| Location | Latitude (°N) | Longitude (°E) | Temperature (°C) [a] | Rainfall (mm) [a] |
|---|---|---|---|---|
| Fukagawa | 43.71 | 142.01 | 16–26/16–26 (14–24/17–27) [b] | 432/243 |
| Miyazaki | 31.82 | 131.41 | 24–32/24–31 (25–32/25–33) | 240/860 |

[a] Average daily minimum and maximum temperatures and total rainfall during the cultivation period in 2016/2017. [b] Figures in parenthesis indicate those during one month after sowing. (https://www.jma.go.jp, accessed on 28 February 2021).

Each strain was pre-incubated on Yeast Mannitol (YM) [14] agar medium at 26 °C for 5–10 days, and a part of the colony was taken into 3 mL of YM liquid medium to adjust $OD_{660}$ at 0.03, then incubated with shaking at 125 rpm for seven days while measuring their $OD_{660}$ at 24-h intervals. All the experiments were done in triplicate.

### 2.2. Effect of Temperature on Competitive Infection of Bradyrhizobium spp. in Soybean

For the competition experiment, each set consisting of three strains from each soil was used as follows: *B. japonicum* Hh 16-9 (Bj11-1), *B. japonicum* Hh 16-25 (Bj11-2), and *B. elkanii* Hk 16-7 (BeL7) from the Fukagawa soil; *B. japonicum* Kh 16-43 (Bj10J-2), *B. japonicum* Kh 16-64 (Bj10J-4), and *B. elkanii* Kh 16-7 (BeL7) from the Miyazaki soil.

The strains were cultured in YM liquid medium with shaking at 25 °C for seven days, then each cell density was adjusted to $10^9$ colony forming unit (CFU)/mL with sterilized distilled water based on OD-CFU/mL correlated linear equations prepared for each strain. Each one milliliter aliquot of the culture was added onto sterilized vermiculite in a 400 mL Leonard jar [15], which was supplemented with sterilized N-free nutrient solution [16]. Three jars were prepared for each treatment. After mixing the inoculated vermiculite thoroughly, three soybean seeds, cv. Orihime (non-Rj) were sown in each Leonard jar and cultivated in a phytotron (LH-220S, NK system, Osaka, Japan) at 20/18 °C and 30/28 °C in 16/8 h (day/night) cycle with an occasional supply of the N-free nutrient solution. The soybean seeds were surface-sterilized prior to sowing with 70% ethanol for 30 s and then with 2.5% NaOCl solution for 3 min [17]. Seedlings were thinned to one plant per jar one week after germination. At three weeks after sowing, the length and weight of the shoot and root were measured, and the number of nodules was counted. Then, nodule composition of the inoculated strains was examined using ten randomly-selected nodules

per plant. Control plants without inoculation were prepared to check contamination, and the experiment was conducted in triplicate. Each nodule was surface sterilized with 70% ethanol for 30 s followed by washing six times with sterilized distilled water, then each nodule was crushed with 200 μL of sterilized MilliQ water for extraction of DNA [18]. The inoculated strain in each nodule was specified by PCR and nucleotide sequence of the 16S-23S rRNA internal transcribed spacer (ITS) region, according to the procedures described previously [7].

*2.3. Statistical Analysis*

Statistical analysis of the soybean growth and nodule compositions of *Bradyrhizobium* spp. were performed using the MSTAT-C 6.1.4 software package [19]. The data were subjected to Duncan's multiple range test after one-way ANOVA.

## 3. Results

*3.1. Effect of Temperature on Growth of Bradyrhizobium spp. Strains in Liquid Culture*

The effects of temperature on the proliferation of the *Bradyrhizobium* spp. strains are presented in Figure 1. The responses to different temperatures varied among the strains. At 15–20 °C, the growth rates of *B. japonicum* Bj11-1 and Bj11-2 were similar and higher than those of *B. elkanii* BeL7 in the Fukagawa strains, and similar growth patterns were observed in *B. japonicum* BjS10J-2 and BjS10J-4, and *B. elkanii* BeL7 in the Miyazaki strains. At 25–35 °C, *B. elkanii* BeL7 proliferated better than the *B. japonicum* strains in the Fukagawa strains, and *B. japonicum* Bj11-1 did not proliferate at 35 °C. Similarly, in the Miyazaki strains, the growth rate of *B. elkanii* BeL7 increased at high temperatures, while those of the *B. japonicum* strains decreased at 30–35 °C, and *B. japonicum* BJS10J-2 did not proliferate at 35 °C.

For each strain, $OD_{660}$ at 5 days of incubation is shown in Figure 2 and normalized as a relative % of $OD_{660}$ to the maximum value in the range of temperatures examined. In the Fukagawa strains, the relative % of Bj11-1 and Bj11-2 were 93–100% at 15–20 °C, while those of BeL7 were 11–13%. The relative % of all strains were more than 80% at 25 °C. At higher temperatures, those of Bj11-1 and Bj11-2 decreased significantly at above 25 and 30 °C, respectively, while those of Bel7 were similar at 25–35 °C. In the Miyawaki strains, BjS10J-2 showed a larger relative % than BjS10j-4 at lower temperatures, and those of BeL7 were less than 20%. At higher temperature, those of BjS10J-2, BjS10j-4, and Bel7 decreased significantly at above 20, 25, and 30 °C, respectively.

*3.2. Effect of Temperature on Growth and Nodule Number of Soybean Inoculated with a Set of Bradyrhizobium spp. Strains*

Effect of temperature on the growth and nodule number of soybean is presented in Figure 3. The shoot and root lengths, and the shoot and root weights were significantly higher at 30/28 °C than 20/18 °C in all treatments except for the root lengths of the soybeans inoculated with Miyazaki strains. While the nodule numbers were not significantly different between the different temperature conditions. The inoculation of the *Bradyrhizobium* spp. strains significantly affected the shoot length and the root weight of soybean at 30/28 °C while the effects were not observed at 20/18 °C. Significant difference in these effects was not present between the Fukagawa and Miyazaki strains. No nodule was recorded in the control plants, indicating that there was no contamination in the experimental procedure.

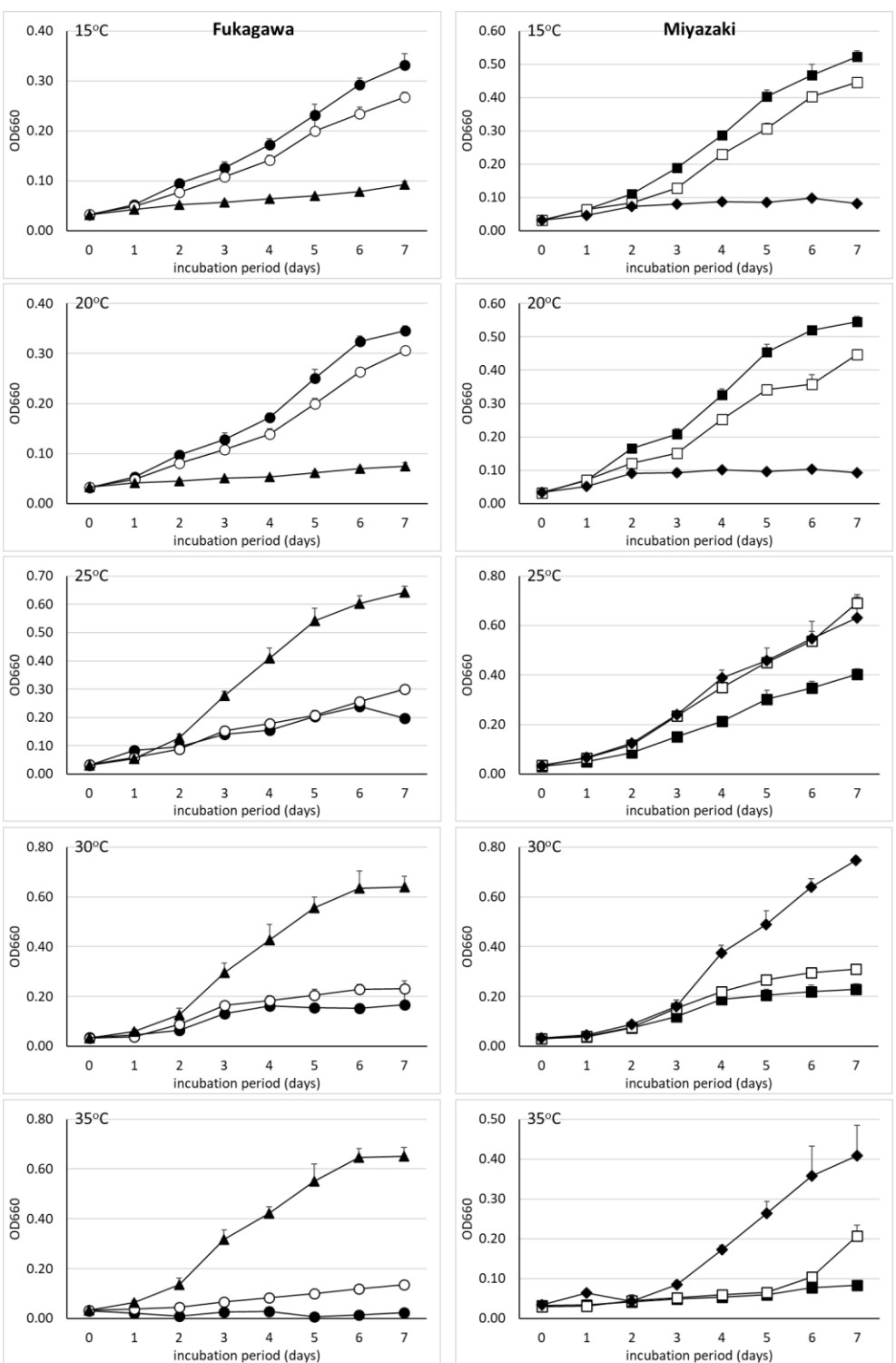

**Figure 1.** Effects of temperature on growth of *Bradyrhizobium* spp. strains in liquid culture. Fukagawa strains: *B. japonicum* Hh 16-9 (Bj11-1) (●), *B. japonicum* Hh 16-25 (Bj11-2) (○), and B. *elkanii* Hk 16-7 (BeL7) (▲); Miyazaki strains: *B. japonicum* Kh 16-43 (Bj10J-2) (■), *B. japonicum* Kh 16-64 (Bj10J-4) (□), and *B. elkanii* Kh 16-7 (BeL7) (◆). The bars represent the standard deviation (*n* = 3).

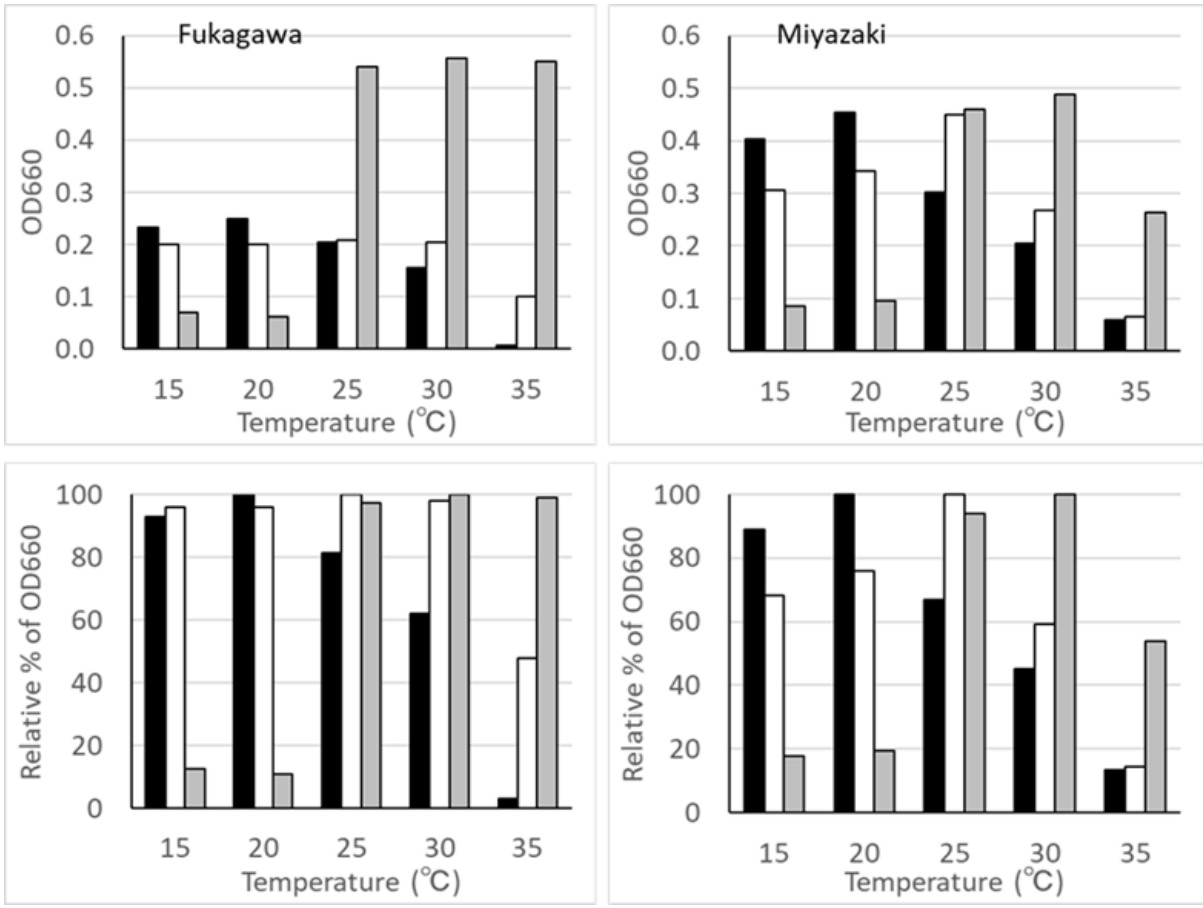

**Figure 2.** Effects of temperature on growth of *Bradyrhizobium* spp. strains in liquid culture. Upper: $OD_{660}$ at 5 days, lower: relative percentage of $OD_{660}$ to maximum for each strain. Fukagawa strains: *B. japonicum* Hh 16-9 (Bj11-1) (■), *B. japonicum* Hh 16-25 (Bj11-2) (□), and B. *elkanii* Hk 16-7 (BeL7) (▧); Miyazaki strains: *B. japonicum* Kh 16-43 (Bj10J-2) (■), *B. japonicum* Kh 16-64 (Bj10J-4) (□), and *B. elkanii* Kh 16-7 (BeL7) (▧).

*3.3. Effect of Temperature on Soybean Nodule Composition of Inoculated Bradyrhizobium spp. Stains*

The relative nodule composition of the inoculated *Bradyrhizobium* spp. strains is presented in Figure 4. Under the competitive conditions for the Fukagawa strains, only Bj11-1 formed the nodules at 20/18 °C, while only BeL7 did at 30/28 °C. For the Miyazaki strains, BjS10J-2 was dominant in the nodules at 20/18 °C with the minor presence of BjS10J-4. At high temperature (30/28 °C) BeL7 was dominant and BjS10J-4 was minor in the nodules. Mixed colonization of nodules with two or three strains in the same nodule would be possible, but minor signals were not visibly observed in the nucleotide chromatogram.

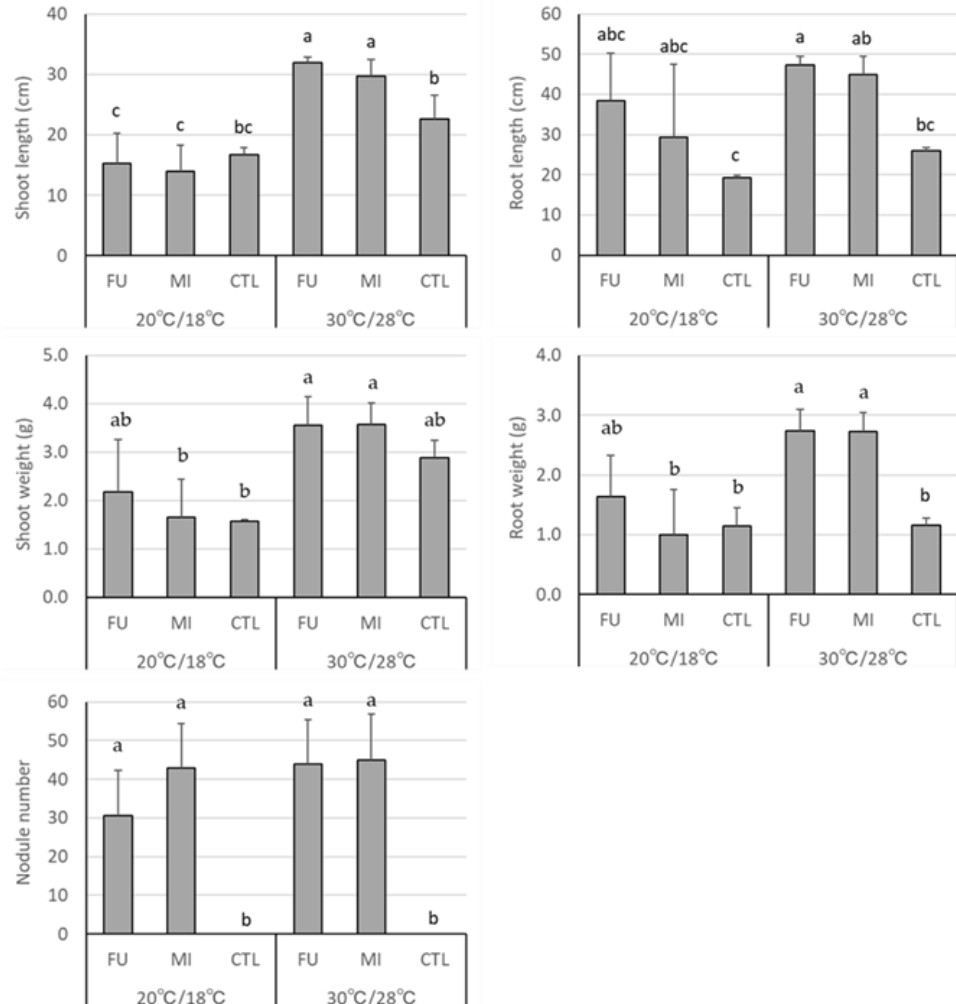

**Figure 3.** Effects of temperature on growth and nodule number of soybean inoculated with a mixture of the *Bradyrhizobium* spp. strains. Soybean was cultivated in a phytotron at 20/18 °C (day/night) and 30/28 °C at 16/8 h cycle. Fukagawa strains: *B. japonicum* Hh 16-9 (Bj11-1), *B. japonicum* Hh 16-25 (Bj11-2), and *B. elkanii* Hk 16-7 (BeL7); Miyazaki strains: *B. japonicum* Kh 16-43 (Bj10J-2), *B. japonicum* Kh 16-64 (Bj10J-4), and *B. elkanii* Kh 16-7 (BeL7); control: no inoculation. The bars represent the standard deviation (*n* = 3) and different letters indicate significant differences at *p* < 0.05 by Duncan's test.

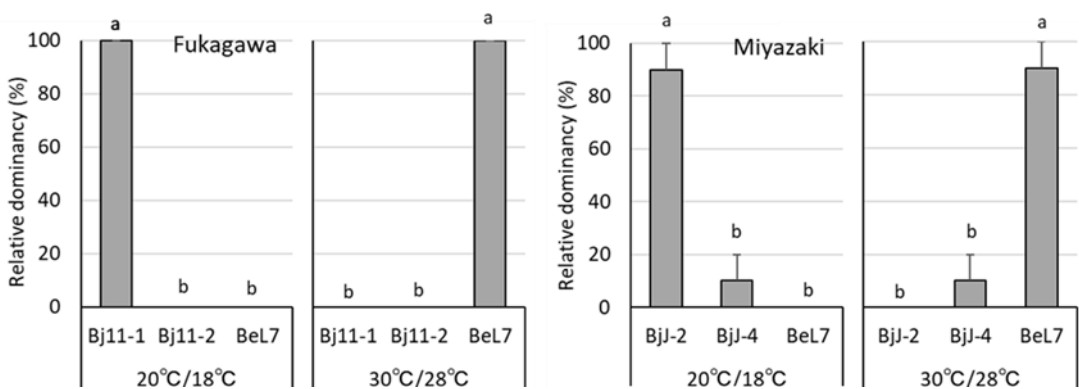

**Figure 4.** Effects of temperature on relative abundancy of inoculated *Bradyrhizobium* spp. strains in soybean. Soybean was cultivated in a phytotron at 20/18 °C (day/night) and 30/28 °C at 16/8 h cycle. Fukagawa strains: *B. japonicum* Hh 16-9 (Bj11-1), *B. japonicum* Hh 16-25 (Bj11-2), and *B. elkanii* Hk 16-7 (BeL7); Miyazaki strains: *B. japonicum* Kh 16-43 (Bj10J-2), *B. japonicum* Kh 16-64 (Bj10J-4), and *B. elkanii* Kh 16-7 (BeL7). The bars represent the standard deviation (*n* = 3) and different letters indicate significant differences at *p* < 0.05 by Duncan's test.

## 4. Discussion

Although the number is limited, a similar temperature-dependent growth tendency in liquid media of the two *Bradyrhizobium* species has been reported previously. Three *B. japonicum* strains grew better at 15 °C than 25 °C, and could not grow at 35 °C, while one *B. elkanii* strain grew better at 25–35 °C than at 15 °C [20]. Kluson et al. [11] also reported that optimum growth of two *B. elkanii* strains was around 25 °C while two *B. japonicum* strains grew best at 20 °C in the range of 20–35 °C. These results suggest that *B. japonicum* and *B. elkanii* have species-specific temperature preference in their proliferations. The tendencies are consistent with the previous results on the latitudinal characteristic nodulation of *B. japonicum* and *B. elkanii* in Japan [7,8], the United States [9], and Nepal [10].

In the infection experiment, we used sterilized vermiculite to simplify the experimental conditions—the same population of the inoculants and elimination of the effects of indigenous soil microorganisms on the competition. Sterilization of soil samples by autoclaving could change its physicochemical conditions. Actually, the population of the inoculated rhizobia decreased in the sterilized Fukagawa soil due to unknown reasons in a preliminary experiment (data not shown). Therefore, we could not use the soil samples in this study.

The better growth of soybean at higher temperature has been reported previously in the similar range of temperatures [11,21,22]. The number of nodules was temperature-independent in this study (Figure 3), while temperature-dependent nodule formation, that is, in this study, the higher temperature, the larger nodule number in the similar temperature range, has been reported when *B. japonicum* strains were inoculated in laboratory experiments [21,23]. In this study, the *B. japonicum* and *B. elkanii* strains were co-inoculated and a different strain was dominant among the inoculated strains in the nodules depending on the temperature (Figure 4), therefore, the nodule number would be dependent on the nodulating properties of the dominant strains in the nodules rather than the temperature.

The high nodule dominancy of *B. elkanii* BeL7 (Hk 16-7 and Kh 16-7) at high temperature (30/28 °C) is presumed to be due to the difference in temperature sensitivity between the *B. japonicum* and *B. elkanii* strains (Figure 2), in addition to the up-regulated expression of *nodC* in *B. elkanii* at high temperature, compared with *B. japonicum* [13]. The temperature-dependent growth properties of the *Bradyrhizobium* spp. strains suggests high nodule dominancy of the *B. japonicum* strains at low temperature (20/18 °C). However, one of the two *B. japonicum* strains for each soil was dominant in the nodules even though their growth properties were similar (Figure 1). Differences in expression levels of nodulation genes and in responses to isoflavones secreted from soybean roots might determine the nodule composition between them. The same temperature-dependent nodule composition; dominancy of B. *japonicum* and B. *elkanii* at low and high temperatures, respectively, has been reported in the other laboratory competitive studies [11–13].

Generally, the composition of soybean rhizobia in field soil has been estimated by nodule composition. Regarding the latitudinal characteristic nodule composition of soybean rhizobia [7–10], competitive inoculation experiments have revealed that the nodule composition is affected by species-specific, temperature-dependent infection and proliferation in soils [11–13]. However, it is uncertain which factor contributes to the temperature-dependent nodule composition.

In our previous study [7], we selected three study locations of different local climatic conditions in Japan, and each soil sample of the study locations was used for soybean cultivation at all the study locations to examine the changes in the nodule compositions under the different local climatic conditions. As a result, we assumed that *B. japonicum* dominantly proliferate in the Fukagawa soil, leading to their dominant nodule composition, because the nodule composition was not affected under warmer climatic conditions in Miyazaki location. To confirm our assumption, the competitive inoculation experiment was conducted using the rhizobial strains isolated from soybean nodules cultivated in Fukagawa soil, and the results showed that *B. japonicum* dominated nodules at lower temperature while *B. elkanii* dominated at higher temperature (Figure 4), supporting our

assumption that *B. japonicum* dominantly proliferate in the Fukagawa soil because the dominancy of *B. elkanii* did not increase at higher temperature in the Miyazaki location.

We also assumed that both *B. japonicum* and *B. elkanii* exist in the Miyazaki soil and the dominant nodule composition of *B. elkanii* is due to their preferred infection because the nodule composition was affected under cooler climatic conditions in Fukagawa location. In the competitive inoculation experiment using the Miyazaki rhizobial strains, *B. japonicum* and *B. elkanii* dominated nodules at lower and higher temperatures, respectively (Figure 4), also supporting our assumption that both *B. japonicum* and *B. elkanii* exist in the Miyazaki soil and their preferred infection determined the nodule composition.

## 5. Conclusions

The experiments performed in the liquid cultures revealed better growth of *B. japonicum* at lower temperatures and *B. elkanii* at higher temperatures, and therefore it can be assumed that the temperature of soil affects rhizobia growth in the rhizosphere and could be a reason for the different competitive properties of *B. japonicum* and *B. elkanii* strains at different temperatures. In addition, competitive infection was suggested between the *B. japonicum* strains.

**Author Contributions:** M.H.R.H. and K.I. conceptualized the study and designed the experiments; M.H.R.H. performed the experiments; A.S. helped to conduct the experiment and in the data analysis; and M.H.R.H. wrote the article, with a substantial contribution from K.I. All authors have read and agreed to the published version of the manuscript.

**Funding:** This research received no external funding.

**Conflicts of Interest:** The authors declare no conflict of interest.

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
