# Peer review of "Growth and Competitive Infection Behaviors of Bradyrhizobium japonicum and Bradyrhizobium elkanii at Different Temperatures"

_horticulturae, doi:10.3390/horticulturae7030041_

Round 1

Reviewer 1 Report

The manuscript "Growth and competitive infection behaviours of Bradyrhizobium japonicum and Bradyrhizobium elkani at different temperatures" by Hafiz et al. describes the characterization of the growth and competitiveness for nodule occupancy of five Bradyrhizobium strains. The Authors show that B. japonicum strains have a preference for a low growth temperature, unlike the B. elkani strain that has a preference for an elevated growth temperature. Similarly, the B. japonicum strains are able to outcompete the B. elkani strains for nodule occupancy at lower temperatures, whereas the B. elkani strain outcompetes the B. japonicum strains at an elevated temperature. Overall, the scientific methods appear appropriate, and the conclusions are generally consistent with the results. I would have liked to see single strain plant inoculation experiments. However, I do not feel these experiments are required. A few specific comments are below.

Line 43: "Bradyrhizobium japonicum" should be shortened to "B. japonicum".

Line 46: I suggest replacing "growing location determine the nodule composition" with "growing location contributes to the nodule composition".

Line 97: Replace "were listed" with "are listed".

It would help the reader to if figure panels were labelled A, B, C, etc.

Lines 192-199: The majority of the results here are not statistically significant according to Figure 3. Either the lack of statistical support should be commented on, or the discussion of these results removed.

Lines 299-301: I don't think it is possible to conclude that "Bj11-1 proliferated dominantly in the Fukagawa soil leading to their nodule dominancy by the more superior competitive infection". It is true that the strain grew faster in liquid culture at lower temperatures and they were more competitive at lower temperatures. However, no evidence is presented that in the plant experiments, Bj11-1 was more abundant in the vermiculite, or that the growth rate phenotype was necessarily linked to the competition for nodule occupancy phenotype. I suggest the Authors consider revising this statement.

Author Response

Thank you for your useful comments on our manuscript. We revised it according to the suggestions as follows:

Line 43: "Bradyrhizobium japonicum" should be shortened to "B. japonicum".

We corrected. (42)

Line 46: I suggest replacing "growing location determine the nodule composition" with "growing location contributes to the nodule composition".

We corrected according to the reviewer’s suggestion. (45-46)

Line 97: Replace "were listed" with "are listed".

We corrected. (96)

It would help the reader to if figure panels were labelled A, B, C, etc.

Labels “Fukagawa” and “Miyazaki” were put on Fig.2.

Lines 192-199: The majority of the results here are not statistically significant according to Figure 3. Either the lack of statistical support should be commented on, or the discussion of these results removed.

We revised according to the reviewer’s suggestion. Significant differences were observed between the different temperature conditions, and the effects of inoculation were significant in high temperature conditions. (191-197)

Lines 299-301: I don't think it is possible to conclude that "Bj11-1 proliferated dominantly in the Fukagawa soil leading to their nodule dominancy by the more superior competitive infection". It is true that the strain grew faster in liquid culture at lower temperatures and they were more competitive at lower temperatures. However, no evidence is presented that in the plant experiments, Bj11-1 was more abundant in the vermiculite, or that the growth rate phenotype was necessarily linked to the competition for nodule occupancy phenotype. I suggest the Authors consider revising this statement.

We revised Conclusion according to the reviewer’s suggestions as follows (299-304):
The experiments performed in the liquid cultures revealed better growth of B. japonicum at lower temperatures and B. elkanii at higher temperatures, and therefore it can be assumed that temperature of soil affects rhizobia growth in rhizosphere and could be a reason for different competitive properties of B. japonicum and B. elkanii strains at different temperatures. In addition, competitive infection was suggested between the B. japonicum strains.

Abstract was also revised accordingly (28-31).

Reviewer 2 Report

1.

Materials and methods: please provide more detail about plant tests: was this experiment conducted once of few times? How many? How many soybean plants were used in each experimental group? How many nodules per experimental group were used for DNA isolation and PCR/sequencing to estimate the percent of colonized nodules for each strain?

2.

Results: have the Authors noticed the mixed colonization of nodules (two or three strains in the same  nodule)? The Authors used lot of rhizobia per plant (1 ml of 109 CFU per 400 ml of plant medium, which gives more than 106 CFU per ml of plant growth medium) so the mixed colonization would be nothing strange…

3.

Results: In my opinion Fig. 2 is unnecessary – Fig 1 provides enough information about growth characteristics of strains at studied temperatures

4.

Conclusions: The Authors wrote “B. japonicum Bj11-1 proliferated dominately in the Fugakawa soil leading to nodule dominancy” or “B. elkani BeL7 […] dominated the nodules due to the higher competitive proliferation and infection”. I think it should be corrected, because the Authors have not performed experiments in Fukagawa soil (but in liquid cultures); similarly, the Authors did not studied the bacterial infection process (they studied nodule occupancy). Therefore it should be stated that – for example – that experiments performed in liquid cultures revealed better growth of B. japonicum at lower temeratures and B. elkanii at higher temperatures, and therefore it can be assumed that temperature of soil affects rhizobia growth in rhizosphere and could be a reason for different competitive properties of B. japonicum and B. elkanii strains at different temperatures.

Author Response

Thank you for your useful comments on our manuscript. We revised it according to the suggestions as follows:

  1. Materials and methods: please provide more detail about plant tests: was this experiment conducted once of few times? How many? How many soybean plants were used in each experimental group? How many nodules per experimental group were used for DNA isolation and PCR/sequencing to estimate the percent of colonized nodules for each strain?

Three pots each containing one plant were set for each group (three plants per group), and ten nodules in one plant (30 nodules per group) were examined for DNA analysis. We revised the method to clarify the procedure. (131-140)

  1. Results: have the Authors noticed the mixed colonization of nodules (two or three strains in the same nodule)? The Authors used lot of rhizobia per plant (1 ml of 109CFU per 400 ml of plant medium, which gives more than 106 CFU per ml of plant growth medium) so the mixed colonization would be nothing strange…

In this study, DNA was directly extracted from the nodules and sequenced (141-146). As minor signals were not visibly observed in the chromatogram, it was suggested that one strain was dominated in each nodule. As the reviewer suggested, the minor mixed colonization would be possible, but it was below the detection level in this study. This information was added in Result 3.3. (217-219)

  1. Results: In my opinion Fig. 2 is unnecessary – Fig 1 provides enough information about growth characteristics of strains at studied temperatures

Fig. 2 summarizes the results in Fig. 1. We would like to present Fig. 2 for better understanding of their relative growth tendencies and optimum temperatures, and for easier comparison among the strains.

  1. Conclusions: The Authors wrote “B. japonicum Bj11-1 proliferated dominately in the Fugakawa soil leading to nodule dominancy” or “B. elkani BeL7 […] dominated the nodules due to the higher competitive proliferation and infection”. I think it should be corrected, because the Authors have not performed experiments in Fukagawa soil (but in liquid cultures); similarly, the Authors did not studied the bacterial infection process (they studied nodule occupancy). Therefore it should be stated that – for example – that experiments performed in liquid cultures revealed better growth of B. japonicum at lower temeratures and B. elkanii at higher temperatures, and therefore it can be assumed that temperature of soil affects rhizobia growth in rhizosphere and could be a reason for different competitive properties of B. japonicum and B. elkanii strains at different temperatures.

We revised Conclusion according to the reviewer’s suggestions as follows (299-304):
The experiments performed in the liquid cultures revealed better growth of B. japonicum at lower temperatures and B. elkanii at higher temperatures, and therefore it can be assumed that temperature of soil affects rhizobia growth in rhizosphere and could be a reason for different competitive properties of B. japonicum and B. elkanii strains at different temperatures. In addition, competitive infection was suggested between the B. japonicum strains.

Abstract was also revised accordingly (28-31).